# Enhancement of Resistive Switching Performance in Hafnium Oxide (HfO_2_) Devices via Sol-Gel Method Stacking Tri-Layer HfO_2_/Al-ZnO/HfO_2_ Structures

**DOI:** 10.3390/nano13010039

**Published:** 2022-12-22

**Authors:** Yuan-Dong Xu, Yan-Ping Jiang, Xin-Gui Tang, Qiu-Xiang Liu, Zhenhua Tang, Wen-Hua Li, Xiao-Bin Guo, Yi-Chun Zhou

**Affiliations:** 1Guangzhou Higher Education Mega Centre, School of Physics and Optoelectronic Engineering, Guangdong University of Technology, Guangzhou 510006, China; 2School of Advanced Materials and Nanotechnology, Xidian University, Xi’an 710126, China; 3Frontier Research Center of Thin Films and Coatings for Device Applications, Academy of Advanced Interdisciplinary Research, Xidian University, Xi’an 710126, China

**Keywords:** HfO_2_, Al-ZnO, sol–gel method, resistive switching, multi-level memory, conductive filament

## Abstract

Resistive random-access memory (RRAM) is a promising candidate for next-generation non-volatile memory. However, due to the random formation and rupture of conductive filaments, RRMS still has disadvantages, such as small storage windows and poor stability. Therefore, the performance of RRAM can be improved by optimizing the formation and rupture of conductive filaments. In this study, a hafnium oxide-/aluminum-doped zinc oxide/hafnium oxide (HfO_2_/Al-ZnO/HfO_2_) tri-layer structure device was prepared using the sol–gel method. The oxygen-rich vacancy Al-ZnO layer was inserted into the HfO_2_ layers. The device had excellent RS properties, such as an excellent switch ratio of 10^4^, retention of 10^4^ s, and multi-level storage capability of six resistance states (one low-resistance state and five high-resistance states) and four resistance states (three low-resistance states and one high-resistance state) which were obtained by controlling stop voltage and compliance current, respectively. Mechanism analysis revealed that the device is dominated by ohmic conduction and space-charge-limited current (SCLC). We believe that the oxygen-rich vacancy concentration of the Al-ZnO insertion layer can improve the formation and rupture behaviors of conductive filaments, thereby enhancing the resistive switching (RS) performance of the device.

## 1. Introduction

Resistive random-access memory (RRAM), a type of next-generation non-volatile memory, has received a lot of attention due to its fast speed, high density, and low power consumption. The RRAM adopts a simple metal–insulator–metal (MIM) structure. At present, a variety of materials can be used to prepare RRAM devices. Transition metal oxides such as SiO_x_ [1,2], Al_2_O_3_ [3,4], ZnO [5], HfO_2_ [6,7], NiO [8], and ZrO [9,10], etc., have been widely explored due to their simple composition and easy preparation. HfO_2_ is a strong competitor for next-generation non-volatile memory materials due to its high dielectric constant of 20–25, bandgap of 5.3–5.7 eV [11], and compatibility with the current complementary metal–oxide–semiconductor (CMOS) technologies. Therefore, HfO_2_-based RRAM has been extensively studied. HfO_2_-based RRAM has a wide range of potential applications, including in combination with other devices, such as 1T1R [12], high-density storage [13], and neuromorphic applications [14]. However, there are still many challenges, such as device scaling [15], forming operation [16], retention [17], and durability are still not as good as DRRM [18], etc. Therefore, further study of HfO_2_-based RRAM is needed to further optimize performance. The origin of resistive switching (RS) in the single layer HfO_2_ thin film depends on the formation and fracture of conductive filaments (CFs) composed of oxygen vacancies in the insulating layer [19]. The formation and distribution of conductive filaments are usually uncontrollable and random. To deal with this problem, a multi-layered structure has been proposed to restrict the formation and random property of CFs, further improving the performance of RRAM devices. Peng et al. [20] studied the HfO_2_/BiFeO_3_/HfO_2_ RRAM devices prepared by inserting the BiFeO_3_ layer into the HfO_2_ layer. The device has a higher switching ratio and multi-level storage capability than the pure HfO2 device, which is attributed to the inserted BiFeO_3_ layer providing sufficient oxygen vacancies. The design of inserting an oxygen-rich vacancy layer plays a key role in regulating the production of CFs. Mahata et al. [21] investigated devices with HfO_2_/Al_2_O_3_/HfO_2_ RRAM. The lack of oxygen vacancies in the Al_2_O_3_ layer between HfO_2_ layers is conducive to the better movement of oxygen vacancies during SET and RESET operations. Oxygen-rich vacancy of the HfO_2_ layers at both ends contributes to the regeneration of CF under SET operation. The specially designed device shows great durability over 10^3^ cycles and multi-stage resistance performance. Neeraj Jain et al. [22] proposed ZnO/HfO_2_ bilayer RRAM devices. The introduction of a HfO_2_ layer can suppress the formation of oxygen defects, thereby improving the on–off ratio. Therefore, one of the most important approaches to improving the performance of thin film devices is the adoption of multi-layer structures. In addition, the thin film preparation method also affects the performance of the film. The HfO_2_-based RRAM devices are prepared by various methods, such as PLD [23], ALD [24,25], RF magnetron sputtering [26], and sol–gel method [27,28], etc. Compared with the ALD and PLD methods required expensive equipment and tedious equipment debugging, sol–gel method has the advantages of easy operation and low cost.

In this work, RRAM devices with tri-layer of HfO_2_/Al-ZnO/HfO_2_ were prepared by sol–gel method, and pure HfO_2_ devices were also prepared as a comparison. The insertion of the oxygen-rich vacancy Al-ZnO layer improves the formation and breakage of CFs, making it possible to obtain a switching ratio greater than 10^4^ and to achieve a multi-stage resistive function by setting a reset stop voltage and compliance current.

## 2. Materials and Methods

### 2.1. Chemicals

In this study, hafnium acetylacetonate C_20_H_28_HfO_8_ (99.7%, Aladdin, Shanghai, China), zinc acetate C_4_H_6_O_4_Zn·2H_2_O (99%, Aladdin, Shanghai, China), Aluminum nitrate nonahydrate Al(NO_3_)_3_·9H_2_O (99%, Aladdin, Shanghai, China), glacial acetic acid CH_3_CO_2_H (AR, Macklin, Shanghai, China), 2-methoxyethanol C_3_H_8_O_2_ (AR, Aladdin, Shanghai, China), acetylacetone C_5_H_8_O_2_ (99%, Aladdin, Shanghai, China) and ethanolamine C_2_H_7_NO (AR, Macklin, Shanghai, China) were used to prepare the films.

### 2.2. Preparation Method

The sol–gel method was used to prepare HfO_2_ and HfO_2_/Al-ZnO/HfO_2_ thin films on ITO/glass substrates. Hafnium acetylacetonate, glacial acetic acid, and 2-methoxyethanol were selected for the preparation of hafnium oxide precursor solutions. Hafnium acetylacetonate was used as the solute, glacial acetic acid and 2-methoxyethanol as the solvent, and acetylacetone as the stabilizer. The calculated hafnium acetylacetonate powder was dissolved in a mixture of glacial acetic acid and 2-methoxyethanol, stirred at 50 °C until dissolved, and then an appropriate amount of acetylacetone was added. The above solution was filtered using filter paper. Then the precursor solution was aged at room temperature for 72 h. Al(NO_3_)_3_·9H_2_O and C_4_H_6_O_4_Zn·2H_2_O were selected as solutes, 2-methoxyethanol as solvent, and ethanolamine as co-solvent for the preparation of the precursor solution of AL-ZnO. The calculated Al(NO_3_)_3_·9H_2_O, C_4_H_6_O_4_Zn·2H_2_O, and appropriate amount of ethanolamine were added to 2-methoxyethanol and stirred at 60 °C for 4 h. After filtration, the precursor solution of aluminum-doped zinc oxide was obtained. Similarly, the solution needed to be aged at room temperature for 72 h. The substrate needed to be cleaned before use to avoid stains on the surface of the substrate affecting the quality of the thin film. The ITO/glass substrate was cleaned in an ultrasonic cleaner for 30 min to remove surface stains, in which anhydrous ethanol and deionized water acted as cleaning agents. The cleaned ITO/glass substrate was dried at low temperature on a drying platform. Appropriate amounts of HfO_2_ precursor solution and Al-ZnO precursor solution were extracted with a syringe. Al-ZnO thin films or HfO_2_ thin films were prepared on ITO/glass substrates on a spin coater on which the rotational speed and duration were set to 1000 rpm for 10 s, and 3000 rpm for 20 s, respectively. After spinning each layer of thin film, a cotton swab was dipped into a trace amount of ethanol to wipe a corner so that it leaked out of the ITO bottom electrode. Then they were dried on a drying platform at 100 °C for 10 min and 300 °C for 20 min. The above process was repeated several times to prepare HfO_2_/Al-ZnO/HfO_2_ thin films and HfO_2_ thin films. Finally, the thin films were annealed for 15 min at 650 °C using a rapid thermal annealing apparatus, under air atmosphere. An ion sputter coater was used to sputter Au top electrode on the thin film to form the MIM structure RS device. The structures of the Au/HfO_2_/ITO device and the Au/HfO_2_/Al-ZnO/HfO_2_/ITO device are shown in Figure 1a,b.

### 2.3. Test and Characterization

The cross sections of the thin films were observed using scanning electron microscopy (SEM) (PHENOM, Shanghai, China). The crystal structure of the thin films was analyzed by grazing incidence X-ray diffraction (GI-XRD) (Bruker, Bremen, Germany). The I-V characteristics of the HfO_2_ and HfO_2_/Al-ZnO/HfO_2_ thin films were tested with a semiconductor parameter analyzer (Keithley 2400) (Solon, OH, USA).

## 3. Results

### 3.1. Structure Analysis

Figure 2a depicts the GI-XRD pattern of HfO_2_/Al-ZnO/HfO_2_ thin film prepared on ITO substrates and the GI-XRD pattern of the ITO substrate. It can be observed that the diffraction peaks at 24.6°, 31.6°, and 50.8° correspond to the (110), (111), and (−221) crystal planes of HfO_2_ in the monoclinic phase, respectively, referring to JCPDS no. 78-0050. The diffraction peaks at 31.7°, 34.4°, and 56.6° correspond to the (100), (002), and (110) crystal planes of Al-ZnO, respectively, referring to JCPDS card no. 36-1451. The Al-ZnO film has a hexagonal wurtzite structure. Figure 2b shows the SEM image of HfO_2_/Al-ZnO/HfO_2_ thin film. It can be observed that there were three demarcated layers on the 200-nm-thick ITO substrate, corresponding to HfO_2_, Al-ZnO, and HfO_2_. The total thickness of the three layers of the film was about 100 nm.

### 3.2. Electrical Performance

In order to investigate the effect of the oxygen-rich vacancy Al-ZnO insertion layer on the resistive performance of HfO_2_, the Au/HfO_2_/ITO, and Au/HfO_2_/Al-ZnO/HfO_2_/ITO devices were tested under the same conditions. Figure 3a shows the formation process of the Au/HfO_2_/ITO and Au/HfO_2_/Al-ZnO/HfO_2_/ITO devices, where the compliance current was set to 1 mA to protect the devices. It can be observed that compared with the Au/HfO_2_/ITO device, the Au/HfO_2_/Al-ZnO/HfO_2_/ITO device had an obvious forming operation, and the forming voltage was 4.4 V. This is attributed to the abundant oxygen vacancy provided by Al-ZnO layer, which accelerates the formation of conductive filaments. Then, the two devices were tested in I–V cycle. The test voltages were cycled in the order of 0 V → V_+_ → 0 V → V_−_. The test results for 100 I–V cycles of the Au/HfO_2_/ITO device and the Au/HfO_2_/Al-ZnO/HfO_2_/ITO device are shown in Figure 3b,c, respectively. Both devices exhibited typical bipolar switching behavior. For the Au/HfO_2_/ITO device, during the voltage increase from 0 V to 3 V, it can be observed that the current increased slowly with the voltage, and the high-resistance state (HRS) switched to a low-resistance state (LRS). Throughout the voltage decrease from 3 V to 0 V, the device was kept as LRS. From 0 V to −3 V, the current initially increased with the voltage and started to decrease gradually when the bias voltage increased to −2.1 V (average of RESET voltage). The device was switched from LRS to HRS during this process. The device remained in the HRS from −3 V to 0 V. It can also be seen that the device performed stably for 100 I–V cycles, with little difference between high- and low-resistance states. For the Au/HfO_2_/Al-ZnO/HfO_2_/ITO device, when the scanning voltage increased from 0 V to 3 V, the current of the device increased sharply at 0.65 V (average of SET voltage), and the resistance state changed from HRS to LRS. From 0 V to −3 V, the current of the device started to drop slowly at −1.34 V (average value of RESETvoltage) and the resistance state changed from LRS to HRS. Compared to the Au/HfO_2_/ITO device, the Au/HfO_2_/Al-ZnO/HfO_2_/ITO device had a more pronounced SET operation, lower RESET voltage, and higher switching ratio. It can be explained by the inserted Al-ZnO layer having a lower dielectric constant than the HfO_2_ layer, which resulted in more oxygen vacancies being generated in the Al-ZnO layer, which facilitated the rapid connection and disconnection of conducting filaments in the resistive layer [29,30]. Figure 3d shows the high/low resistance states of both devices at 100 cycles, read at −0.45 V. The switching ratio of the Au/HfO_2_/ITO device is 2, while that of the Au/HfO_2_/Al-ZnO/HfO_2_/ITO device is 10^2^. The high-/low-resistance state showed good stability with no significant downward trend. Figure 3e shows the cumulative probability plots of HRS and LRS for both devices. The coefficients of variation (defined as the standard deviation δ divided by the mean μ) of the high- and low-resistive state resistances were calculated, respectively[31]. The coefficients of variation of the HRS and LRS of the Au/HfO_2_/ITO device were 0.09 and 0.08, respectively, while those of the Au/HfO_2_/Al-ZnO/HfO_2_/ITO device were 0.23 and 0.04, respectively. It can be found that the HRS and LRS of both devices exhibited good stability. Compared with the Au/HfO_2_/ITO device, the Au/HfO_2_/Al-ZnO/HfO_2_/ITO device had a smaller coefficient of variation and better stability in the LRS, while the HRS fluctuated slightly. In order to study the data retention of the devices, the two devices were subjected to retention tests at 85 °C for 10^4^ s. Neither device showed significant decline, as shown in Figure 3f.

Furthermore, the I–V characteristics of the Au/HfO_2_/Al-ZnO/HfO_2_/ITO devices were tested at different operating voltages, where excellent switching ratios were obtained at operating voltages from −5 V to 5 V. As shown in Figure 4a, the LRS was relatively stable, while the HRS showed slight fluctuations. This may be attributed to the presence of multiple conductive filaments [32]. Its resistance state durability is shown in Figure 4b, which reads 0.25 V. The switching ratio is 10^4^ and there is no trend of decay for 100 cycles. Figure 4c shows the corresponding cumulative probability plots for the HRS and LRS. The coefficients of variation for the LRS and HRS are 0.125 and 0.609, respectively. Figure 4f shows that the retention characteristic of high and low resistance of the device has no obvious decline. In addition, the pulse endurance of the device is shown in Appendix A of the Appendix A. The resistive switching of device failed after 120 pulse cycles, which was attributed to the fact that there were not enough oxygen ions to compound with the oxygen vacancy, causing the device to remain at LRS. This comparative experiment demonstrates that the design of multi-layer resistive layers is beneficial for improving the switching ratio of the device. The performance of this work compared to other HfO_2_-based RRAM is shown in Table 1.

RRAM can exhibit multi-level HRS and LRS by controlling the compliance current (I_CC_) or reset stop voltage during switching, which is very advantageous for implementing multi-level storage. Usually, compliance current affects the resistance distribution of the switching process. In the I–V graph of the Au/HfO_2_/Al-ZnO/HfO_2_/ITO device at various I_CC_ (5 mA, 8 mA, and 10 mA), the LRS resistance decreases with the increase of I_CC_, as shown in Figure 5a. The reason for this can be attributed to the fact that the diameters of the conductive filaments can be adjusted by setting the I_CC_ [40]. Figure 5b shows the resistance values of the high- and low-resistance states at different compliance currents with a reading of 0.3 V (50 cycles of each). The high- and low-resistance states can be clearly distinguished, and the three different LRSs can be clearly identified. The LRS exhibited a reliable stability and the HRS fluctuated slightly. The I_CC_ increased from 5 mA to 10 mA and the average values of the corresponding switching ratios were 2.8 × 10^2^, 2.8 × 10^3^, and 4.4 × 10^3^, respectively. Further, the I–V plot of Au/HfO_2_/Al-ZnO/HfO_2_/ITO device tested at different reset stop voltages (−2 V, −2.3 V, −2.5 V, −3.5 V, and −4 V) with I_CC_ set at 10 mA, is shown in Figure 5c. It can be observed that the current in the high-resistance state decreased as the reset stop voltage increased, witch can be explained by the reset process of the device being the recombination of oxygen ions with oxygen vacancies [41], and the increase of the applied negative pressure enhancing the oxygen exchange reaction, which led to a decrease in the HRS current [42]. The stability of the high- and low-resistance states corresponding to different stop voltages is shown in Figure 5d, with a reading of −0.5 V (40 cycles of each). The LRS resistance was basically constant, and the HRS resistance increased with the increase in the stop voltage. There are five high resistance states that can be clearly distinguished. Different stop voltages correspond to average values of switching ratios of 16, 49, 1.4 × 10^2^, 2.6 × 10^3^, and 2.5 × 10^4^, respectively. As shown in Figure 5e,f, the retention characteristics of different resistances were clearly differentiated and there was no obvious sign of decline. The above multi-level RS characteristics suggest that the Au/HfO_2_/Al-ZnO/HfO_2_/ITO device can be applied to multi-level storage.

### 3.3. Mechanism Discussion

The I-V characteristic curve of the Au/HfO_2_/Al-ZnO/HfO_2_/ITO device were redrawn into double logarithmic coordinates and the Log(I)–Log(V) curves were fitted linearly to explore the switching mechanism of the device. As shown in Figure 6a, in the process of 0–4 V voltage scanning, the slope of the low voltage region was 0.97, indicating that this region was mainly dominated by Ohmic conduction mechanism [43]. This indicates that thermally generated free carriers were predominant in this region. In this case, the current density due to ohmic conduction is expressed as [44]:*J_Ohm_ = qn_0_μ(V/d)*(1)
where *q* is the elementary charge, *n*_0_ is the free carrier density, *μ* is the mobility, *V* is the applied voltage, and *d* is the thickness of the thin film. As the voltage increased, the slope of the Log(I)–Log(v) curve was 1.96, indicating that the region was consistent with Child’s Law [45]. Then the current increased dramatically and the resistance state changed from HRS to LRS, in which it completed the SET operation. These results imply that HRS in positive pressure regions was dominated by spatial charge-limiting current (SCLC) mechanisms [46,47]. At low voltage, the electrons injected from the outside were trapped in the dielectric, and as the voltage increased, the number of injected electrons - increased. When all the traps were filled, all the injected electrons became free electrons, and the current increased dramatically and completed the resistance state transition. The current generated by SCLC can be expressed as[48]:*J_SCLC_* = (9/8)*μεθ*(*V*^2^/*d*^3^)(2)
where *μ* is the electron mobility, *ε* is the permittivity of the film, and *θ* is the ratio of free and trapped charge. The resistance state remained LRS during the scan voltage drop from 4 V to 0 V. The slope of the high voltage region was 1.79, meaning that a large number of traps were still filled with electrons. As the voltage decreased, the slope was 1.29, which can be explained by the ohmic conduction mechanism. The slope was 1.64 when the scan voltage increased from 0 V to −4 V in the reverse bias region, showing a SCLC mechanism, as shown in Figure 6b. Then the current started to decrease slowly and the resistance switched from LRS to HRS to complete the RESET operation. During the voltage scan of −4–0 V, the slopes of the three segments were 2.91, 1.70, and 1.09 as the voltage decreased, indicating that the SCLC mechanism was dominant in this segment. The resistance state always remained as HRS during this process. Through the above analysis, we can conclude that the conduction mechanism of Au/HfO_2_/Al-ZnO/HfO_2_/ITO memory devices was dominated by ohmic conduction and SCLC.

In order to better understand the RS process of the device, we propose a reasonable model to explain the operating mechanism of the Au/HfO_2_/Al-ZnO/HfO_2_/ITO device. In the initial state, oxygen vacancies are randomly distributed in the resistive layer, in which the Al-ZnO layer should have more oxygen vacancies than the HfO_2_ layer, as shown in Figure 7a. The main reason is that the dielectric constant of Al-ZnO is much lower than that of HfO_2_. The low dielectric constant of Al-ZnO thin film leads to easy breakage of Zn–O bonds and therefore more oxygen vacancies are generated within the Al-ZnO layer. When a positive bias voltage is applied to the Au top electrode (TE), oxygen ions move away from their sites toward the Au TE under the action of the electric field and create oxygen vacancies at their sites. At the same time, oxygen vacancies move to the bottom electrode (BE) of the ITO and gradually accumulate near the ITO BE, as shown in Figure 7b. With the gradual increase of the applied bias voltage, a conductive filament consisting of oxygen vacancies is formed between the ITO BE and the Au TE. The current increases instantaneously, as shown in Figure 7c. At this point, the memory device switches from HRS to LRS, generating a set operation. The thickness of the formed conductive filaments varies due to the different oxygen vacancy content of the different thin-film layers [49]. From this, we can speculate that the conductive filament diameter of the Al-ZnO layer is larger than that in the HfO_2_ layer. The interface layer consisting of different dielectrics forms the weakest conductive filament [21]. The conductive filament is easy to connect and rupture at the interface [50]. In contrast, when a negative bias voltage is applied to the Au TE, the oxygen ions stored at the interface move towards ITO BE and recombine with the oxygen vacancy under the applied bias. Therefore, the conductive filament will rupture at the Au/HfO_2_, HfO_2_/Al-ZnO, and Al-ZnO/HfO_2_ interfaces, as shown in Figure 7d. The device was switched from the LRS to the HRS, and the undissolved conducting filament would be used as the starting point for the next trigger state switch.

## 4. Conclusions

In conclusion, we prepared multi-layer HfO_2_/Al-ZnO/HfO_2_ thin films using the sol–gel method. By inserting an Al-ZnO layer with oxygen-rich vacancies into the HfO_2_ layer, the connection and rupture of the conducting filaments were improved, resulting in an improved switching ratio of the devices.

The results show that the device had an excellent switching ratio of 10^4^ and good stability. After 100 cycles, the switching ratio had no obvious downward trend. Meanwhile, six resistance states (five HRS and one LRS) and four resistance states (three LRS and one HRS) could be obtained by controlling the stop voltage and setting the I_CC_, respectively. This has important implications for the realization of RRAM with multi-level storage. The resistance mechanism of the device was studied, and it was found that the conduction mechanism of the device was dominated by the SCLC mechanism and Ohmic conduction mechanism. This work is of great significance for further improving the performance of HfO_2_-based RRAM.

## Figures and Tables

**Figure 1 nanomaterials-13-00039-f001:**
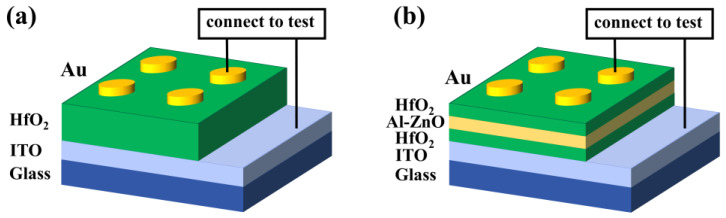
(**a**) Structure of Au/HfO_2_/ITO device and (**b**) Au/HfO_2_/Al-ZnO/HfO_2_/ITO device.

**Figure 2 nanomaterials-13-00039-f002:**
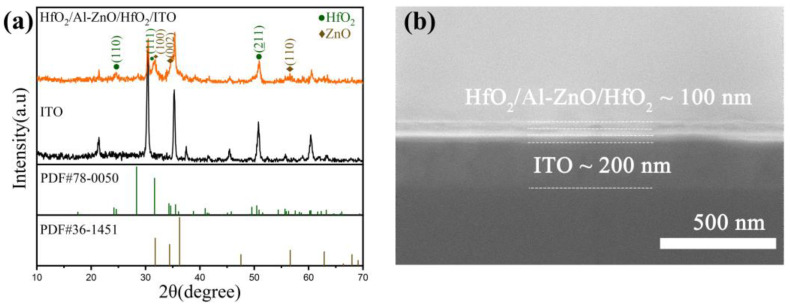
(**a**) XRD patterns of ITO substrate and HfO_2_/Al-ZnO/HfO_2_/ITO thin film, and (**b**) SEM image of HfO_2_/Al-ZnO/HfO_2_/ITO.

**Figure 3 nanomaterials-13-00039-f003:**
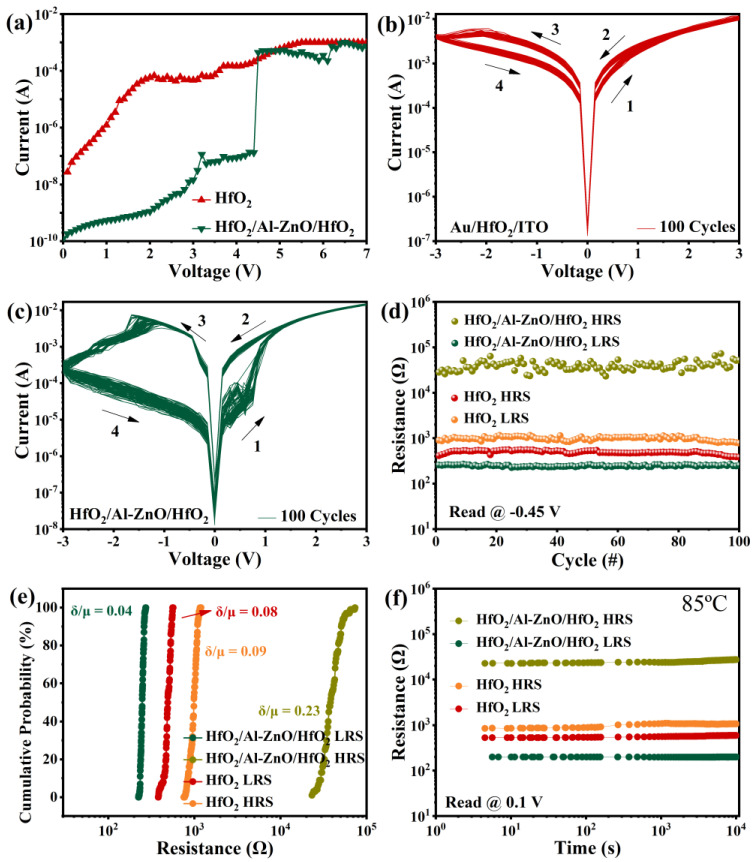
(**a**) Forming operations; (**b**) I–V characteristics of Au/HfO_2_/ITO memory device; (**c**) Au/HfO_2_/Al-ZnO/HfO_2_/ITO memory devices; (**d**) high-/low-resistance states of Au/HfO_2_/ITO and Au/HfO_2_/Al-ZnO/HfO_2_/ITO devices for 100 cycles, read at −0.45 V; (**e**) cumulative probability distribution of the LRS/HRS resistance of the Au/HfO_2_/ITO and Au/HfO_2_/Al-ZnO/HfO_2_/ITO devices; and (**f**) retention of Au/HfO_2_/ITO device and Au/HfO_2_/Al-ZnO/HfO_2_/ITO device, read at 0.1 V.

**Figure 4 nanomaterials-13-00039-f004:**
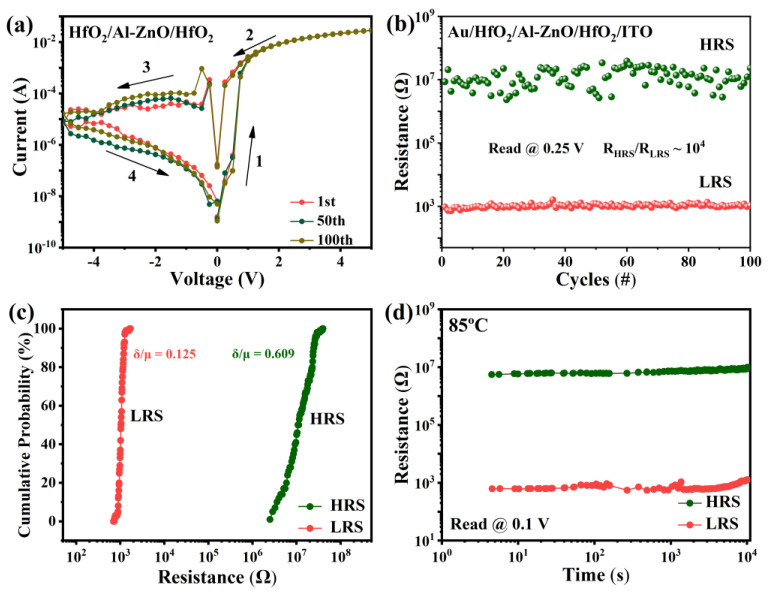
(**a**) I–V characteristics of Au/HfO_2_/Al-ZnO/HfO_2_/ITO device at operating voltage (−5–5 V); (**b**) 100 cycles of high-/low-resistance state resistance distribution, read at 0.25 V; (**c**) cumulative distribution of HRS and LRS; and (**d**) retention of resistance states, read at 0.1 V.

**Figure 5 nanomaterials-13-00039-f005:**
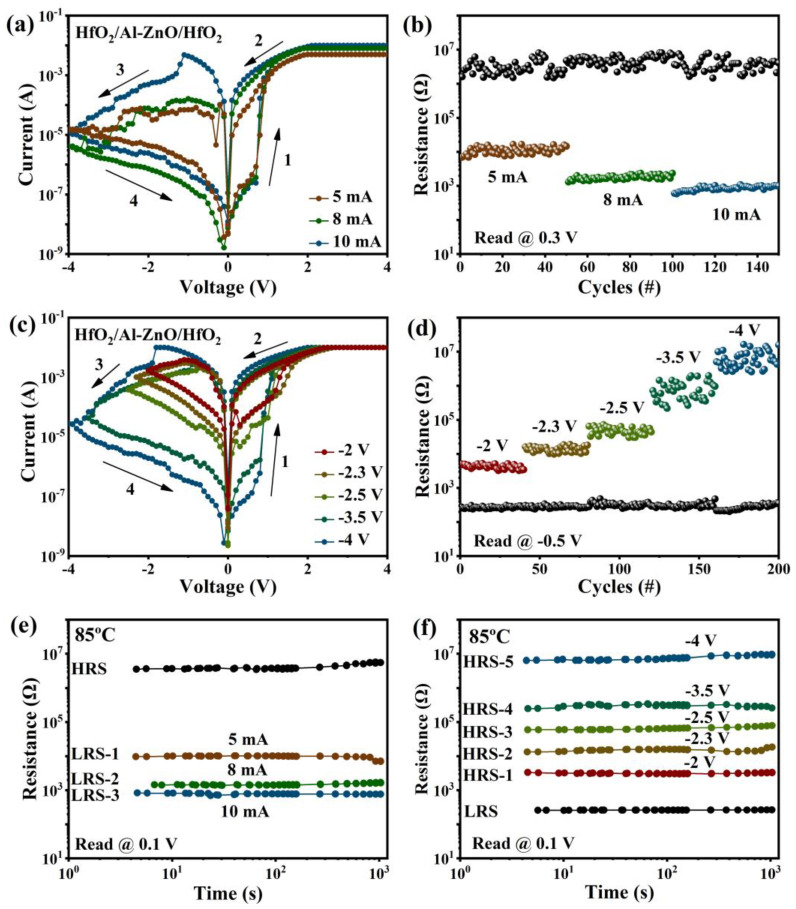
Multi-level storage characteristics of Au/HfO_2_/Al-ZnO/HfO_2_/ITO memory device: (**a**) I–V characteristics of different compliance currents (5 mA, 8 mA, and 10 mA); (**b**) resistance distribution of high-/low-resistance states corresponding to different compliance currents (5 mA, 8 mA, and 10 mA), read at 0.3 V; (**c**) I–V characteristics at different stop voltages (−2 V, −2.3 V, −2.5 V, −3.5 V and −4 V); and (**d**) high-/low-resistance state resistance distribution corresponding to different reset stop voltages (−2 V, −2.3 V, −2.5 V, −3.5 V, and −4 V), read at −0.5 V; (**e**) retention of resistance states under different compliance currents; and (**f**) retention of resistance states under different stop voltages.

**Figure 6 nanomaterials-13-00039-f006:**
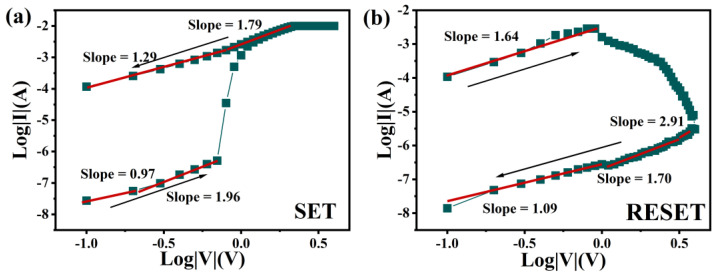
Log(V)–Log(I) fitting diagram of Au/HfO_2_/Al-ZnO/HfO_2_/ITO device: (**a**) set operation, and (**b**) reset operation.

**Figure 7 nanomaterials-13-00039-f007:**
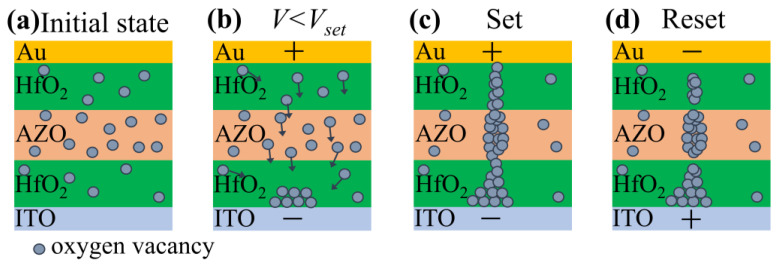
(**a**) Conduction model of the Au/HfO_2_/Al-ZnO/HfO_2_/ITO device with oxygen vacancies distributed in the insulating layer in the initial state, (**b**) oxygen vacancies move towards the ITO bottom electrode, (**c**) applied voltage greater than *V_set_* on the top electrode and oxygen vacancies form conductive filaments, and (**d**) applied negative voltage to the electrode and broken conductive filament.

**Table 1 nanomaterials-13-00039-t001:** Comparison of HfO_2_-based RRAM.

Structure of Device	V_Forming_ [V]	V_set_ [V]	V_reset_ [V]	HRS/LRS	Endurance	Retention [s @ °C]	
Ti/HfO_2_/Pt	Free	0.88	−0.89	11.4	10^4^	10^4^ @ 100	[7]
Au/HfO_2−x_/Pt	2.3	1	-	100	10^3^	10^4^ @ 85	[33]
Pt/HfO_2_/TiO_2_/HfO_2_/Pt	3.2	1.5	−0.5	<100	10^2^	10^4^ @ -	[34]
Pt/HfO_2_/TiO_2_/ITO	3.1	1.6	−1.5	>10	10^2^	10^4^ @ 85	[31]
ITO/S:HfO_x_/TiN	0.85	0.11	−0.15	90	10^6^	10^4^ @ 85	[35]
Pt/HfO_2_/In_2_O_3_/TiN	-	0.59	−0.52	15	10^7^	10^4^ @ 85	[36]
Pt/HfO_2_/BiFeO_3_/HfO_2_/TiN	5.2	-	-	104	10^6^	10^4^ @ 85	[14]
Pt/HfO_2_/TiN	Free	0.67	−0.66	112	10^4^	10^4^ @ 85	[16]
In/HfO_2_/TiN	−5.3	-	-	10^7^	10^2^	10^4^ @ 85	[37]
Al/Ti/HfO_2_/Pt/Ti	2	1.05	-	10^4^	10^2^	10^4^ @ -	[38]
Cu/HfO_2_/p^++^Si	-	3	−0.5–−1	10^4^	-	10^4^ @ -	[39]
This work	4.4	0.65	-1.34	10^4^	10^2^	10^4^ @ 85	

## Data Availability

The data presented in this study are available on request from the corresponding author.

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
