# Peer review of "Enhancement of Resistive Switching Performance in Hafnium Oxide (HfO2) Devices via Sol-Gel Method Stacking Tri-Layer HfO2/Al-ZnO/HfO2 Structures"

_nanomaterials, 2022, doi:10.3390/nano13010039_

Round 1

Reviewer 1 Report

The current manuscript is discussing the HfO2-based RRAM device with Al-ZnO. At this point, this study is not a complete one. Further clarifications are necessary. The comments are as bellow 

1.       The impact of switching materials in RRAM is already reported with high performance and reviewed several times. (Electronics 9 (6), 1029, 2020; 10.1109/LED.2013.2251857). Apart from the Al-ZnO layer where is the performance improvement in this paper?

2.       A 100 nm thick film is way too thick. What will be the minimum working thickness for this structure?

3.       How the device area scaling can impact performance?

4.       Forming performance is missing. Additionally please explain the switching mechanism schematically.

5.       Authors have misconceptions about non-volatile memory. Without 85oC data retention, non-volatile memory can’t exist. Please provide full endurance behavior and high-temperature retention before claiming non-volatile memory.

6. The review of HfO2-based RRAM is very poor. Especially recent reports on the prospect and challenges of HfO2 is missing. Please improve the reference section with the suggested references.

Author Response

Response Letter (nanomaterials-2042957)

Manuscript ID: nanomaterials-2042957

Title: Enhancement of Resistive Switching Performance in Hafnium Oxide (HfO2) Devices via Sol-gel Method Stacking Tri-layer HfO2/Al-ZnO/HfO2 Structures

Dear editors and reviewers,

Thank you very much for your insightful comments to our paper submitted to Nanomaterials. These comments are very helpful to revise the manuscript accordingly. Based on these comments, we have supplemented the relevant experiments and tests, added more sufficient explanations for the parts that need to be explained, and the corresponding modifications are shown in the manuscript. We reply to your comments and resubmitted to you the revised manuscript, we sincerely hope this revision can make this manuscript acceptable.

Below are the comments by you and our replies. All changes have been highlighted in red through the manuscript.

Comments from the editors and reviewers:

  •  

Referee 1:

Comments and Suggestions for Authors

The current manuscript is discussing the HfO2-based RRAM device with Al-ZnO. At this point, this study is not a complete one. Further clarifications are necessary. The comments are as bellow 

Firstly, thank you very much for your valuable comments and suggestions. According to these comments, the modification but necessary changes have been made, which have been highlighted in red through the revised version and explained one by one as follows:

Comment1: The impact of switching materials in RRAM is already reported with high performance and reviewed several times. (Electronics 9 (6), 1029, 2020; 10.1109/LED.2013.2251857). Apart from the Al-ZnO layer where is the performance improvement in this paper?

Reply: We appreciate the reviewer’s comment. In addition to Al-ZnO, the sol-gel method, which is low-cost and suitable for large area preparation, was used to prepare HfO2-based multi-layer films. HfO2-based thin films prepared by this method are rarely reported at present. We try to prepare HfO2-based thin films by a low-cost method. Thanks again for your comment.

Comment2: A 100 nm thick film is way too thick. What will be the minimum working thickness for this structure?

Reply:  The thickness of the films is partly due to the sol-gel preparation method. The thickness of the film prepared by sol-gel method is generally about 100 nm, refer to the reference [1,2]. The thickness of the film shown in the paper is the thickness for optimal performance. At the beginning of the experiment, we designed HfO2/Al-ZnO/HfO2 devices with different thickness of Al-ZnO insertion layer, and found that devices with thinner Al-ZnO layer could not obtain stable I-V cycle. Only when the thickness of Al-ZnO layer is increased to a certain thickness, HfO2/Al-ZnO/HfO2 devices can obtain stable resistive switching performance.

Comment3: How the device area scaling can impact performance?

Reply:  Thank you very much for your comments. At present, the laboratory does not have the technical equipment to prepare the precise area scaling sample. Therefore, experiments related to area scaling cannot be carried out. If experimental conditions are available later, the area scaling of the device will be studied. Thanks again for your comment.

Comment4: Forming performance is missing. Additionally please explain the switching mechanism schematically.

Reply:  We agree with the reviewer’s comment. According to your suggestion, we have added relevant experiments. Test results for the Forming operation of both devices have been added to the manuscript. (See Figure 3a on page 6) The different Forming operations of the two devices are explained in the manuscript. (See row 156-162 on page 4) Thanks again for your comment.

Comment5: Authors have misconceptions about non-volatile memory. Without 85oC data retention, non-volatile memory can’t exist. Please provide full endurance behavior and high-temperature retention before claiming non-volatile memory.

Reply:  We agree with the reviewer’s comment. According to your suggestion, we have added relevant experiments. Relevant data retention test results have been added to the manuscript. (See Figure 3f, Figure 4d, Figure 5e and Figure 5f) In addition, in the durability test of the device, we could not get stable data, so the relevant durability of the device was not added in the manuscript. Thanks again for your comment.

Comment6: The review of HfO2-based RRAM is very poor. Especially recent reports on the prospect and challenges of HfO2 is missing. Please improve the reference section with the suggested references.

Reply: Thanks for your constructive advice. According to your suggestion, we have added a review of HfO2-based RRAM in the introduction to the manuscript. We have carefully read the article you recommended and listed it as a reference in the manuscript. (See Reference [14]) Thank you very much for your comments.

Thank you for your attentions. Your comments are helpful for our future work. 

Yours Sincerely,

 Yuan-Dong Xu

  1. Hsu, C.-C.; Chuang, H.; Jhang, W.-C. Annealing effect on forming-free bipolar resistive switching characteristics of sol-gel WOxresistive memories with Al conductive bridges. J. Alloys Compd. 2021, 882, 160758.
  2. Liu, C.-F.; Tang, X.-G.; Wang, L.-Q.; Tang, H.; Jiang, Y.-P.; Liu, Q.-X.; Li, W.-H.; Tang, Z.-H. Resistive switching characteristics of HfO2thin films on mica substrates prepared by Sol-Gel process. Nanomaterials 2019, 9, 1124.

Reviewer 2 Report

1-    The abstract is too short and is not representing the important results of this study. Rewrite this part.

2-    The keywords are less than 6, add one more keyword. Also, change “thin film” with another keyword.

3-    hfO2 is not explained in the abstract and in the title. Use the full form first, then use the abbreviation.

4-     Separate Materials and Methods parts. The two parts are merged in this manuscript.

5-    What is the role of HfO2 and why did the authors select HfO2 to use?

6-    The authors can use the following references for improving the discussion in the manuscript:

Green synthesis of Mg0. 99 Zn0. 01O nanoparticles for the fabrication of κ-Carrageenan/NaCMC hydrogel in order to deliver catechin. Polymers12(4), 861.

Author Response

Response Letter (nanomaterials-2042957)

Manuscript ID: nanomaterials-2042957

Title: Enhancement of Resistive Switching Performance in Hafnium Oxide (HfO2) Devices via Sol-gel Method Stacking Tri-layer HfO2/Al-ZnO/HfO2 Structures

Dear editors and reviewers,

Thank you very much for your insightful comments to our paper submitted to Nanomaterials. These comments are very helpful to revise the manuscript accordingly. Based on these comments, we have supplemented the relevant experiments and tests, added more sufficient explanations for the parts that need to be explained, and the corresponding modifications are shown in the manuscript. We reply to your comments and resubmitted to you the revised manuscript, we sincerely hope this revision can make this manuscript acceptable.

Below are the comments by you and our replies. All changes have been highlighted in red through the manuscript.

Comments from the editors and reviewers:

Referee 2:

Comments and Suggestions for Authors

Reply: Thank you for your valuable comments and suggestions. According to these comments, the modification but necessary changes have been made, which have been highlighted in red through the revised version and explained one by one as follows:

Comment1: The abstract is too short and is not representing the important results of this study. Rewrite this part.

Reply: Thank you very much for your valuable comments and suggestion. According to your suggestion, we have rewritten the abstract. (See row 15-29 on page 1) Thank you again for your suggestion.

Comment2: The keywords are less than 6, add one more keyword. Also, change “thin film” with another keyword.

Reply: Thanks for your constructive advice. We have revised and supplemented the keywords in the manuscript. (See row 39 and 40 on page 1) Thank you again for your suggestion.

Comment3: HfO2 is not explained in the abstract and in the title. Use the full form first, then use the abbreviation.

Reply: Thanks for your constructive advice. The full form of HfO2 has been added to the abstract and the title. Thank you again for your suggestion. (See the title and row 19 on page 1)

Comment4: Separate Materials and Methods parts. The two parts are merged in this manuscript.

Reply: Thank you very much for your comments. We have divided Materials and Methods into Chemicals, Preparation methods and Test and Characterization. (See row 88 on page 2 and row 95 and 130 on page 3) Thanks again for your comment.

Comment5: What is the role of HfO2 and why did the authors select HfO2 to use?

Reply: HfO2 is prepared between two electrodes as a resistive dielectric layer to form a RRAM device with metal-insulator-metal (MIM) structure. Under different bias voltages, different resistive states will be generated. The aim of the experiment is to improve the resistive switching of HfO2 by inserting Al-ZnO into HfO2. We chose to study HfO2 because HfO2 is a high dielectric constant and compatible with CMOS processes, and therefore has the potential for commercial production. The reason why AL-ZnO is chosen as the material of the insertion layer is that Al-ZnO has low dielectric constant, so it will generate more oxygen vacancies than HfO2, so as to achieve the purpose of improving the conductive filament and improve the performance of HfO2-based devices. Thank you very much for your comments.

Comment6: The authors can use the following references for improving the discussion in the manuscript:

Green synthesis of Mg0. 99 Zn0. 01O nanoparticles for the fabrication of κ-Carrageenan/NaCMC hydrogel in order to deliver catechin. Polymers, 12(4), 861.

Reply: Thanks for your constructive advice. According to your suggestion, we have listed this article as a reference in the manuscript. (See references [26]) Thank you very much for your comments.

Thank you for your attentions. Your comments are helpful for our future work. 

Yours Sincerely,

Yuan-Dong Xu

Round 2

Reviewer 1 Report

The authors tried to improve the manuscript but ignored several comments. Please address the comments properly.

1.       Apart from the Al-ZnO layer where is the performance improvement in this paper? (same as last time). Please provide a comparison table with other hfO2-based RRAM/CBRAM devices within the last 5 years. This can provide the readers with a fair understanding of the development of this particular material system.

2.       Although I can consider the thicker film due to the sol-gel method. However, the question is for mass production using other methods like PVD and all, what will be the thickness limitation of the same stack?

3.       Why in your case pure HfO2 show high leakage and showing poor switching (F3b)? But other groups reported much better sol-gel HfO2-based RRAM. (Scientific Reports volume 9, Article number: 9983 (2019)). Please comment on that.

4.       Authors show the switching mechanism in F7. Now the question is in all switching is happening in HfO2 (F7d), then why do you need a complicated stack design? What is the purpose of each layer please explain.

5.       As mentioned by the authors “In addition, in the durability test of the device, we could not get stable data, so the relevant durability of the device was not added in the manuscript.” You can not find the expected data always. But you must show what you have and explain it. There is always scope for improvement.

6. In the previous review, I mentioned that “The review of HfO2-based RRAM is very poor. Especially recent reports on the prospect and challenges of HfO2 are missing.” It is still not done. Reference 14 is almost 10 years old now. Try to include review papers (Small, 2107575, 2022) on HfO2 which will give a border view to readers.

Author Response

Response Letter (nanomaterials-2042957)

Manuscript ID: nanomaterials-2042957

Title: Enhancement of Resistive Switching Performance in Hafnium Oxide (HfO2) Devices via Sol-gel Method Stacking Tri-layer HfO2/Al-ZnO/HfO2 Structures

Dear editors and reviewers,

Thank you very much for your insightful comments to our paper submitted to Nanomaterials. These comments are very helpful to revise the manuscript accordingly. Based on these comments, we have supplemented the relevant experiments and tests, added more sufficient explanations for the parts that need to be explained, and the corresponding modifications are shown in the manuscript. We reply to your comments and resubmitted to you the revised manuscript, we sincerely hope this revision can make this manuscript acceptable.

Below are the comments by you and our replies. All changes have been highlighted in red through the manuscript.

Comments from the editors and reviewers:

  •  

Referee 1:

Comments and Suggestions for Authors

The authors tried to improve the manuscript but ignored several comments. Please address the comments properly.

Firstly, thank you very much for your valuable comments and suggestions. According to these comments, the modification but necessary changes have been made, which have been highlighted in red through the revised version and explained one by one as follows:

Comment1: Apart from the Al-ZnO layer where is the performance improvement in this paper? (same as last time). Please provide a comparison table with other hfO2-based RRAM/CBRAM devices within the last 5 years. This can provide the readers with a fair understanding of the development of this particular material system.

Reply: Thank you very much for your comments. According to your suggestion, we have added the table of comparison between this work and other HfO2 RRAM in the manuscript. Thanks again for your comment. (See Table 1)

Comment2: Although I can consider the thicker film due to the sol-gel method. However, the question is for mass production using other methods like PVD and all, what will be the thickness limitation of the same stack?

Reply:  We appreciate the reviewer’s comment. We did not do a detailed study of the device thickness, we just picked out the best performance of the insertion layer thickness, and then compared with pure HfO2 devices. So there is no ability to respond to questions about device thickness limits. We are very sorry about this. It is our negligence in the design of the whole experimental scheme. We will design and study the thickness of HfO2 prepared by sol-gel method in the future. Again, our apologies.

Comment3: Why in your case pure HfO2 show high leakage and showing poor switching (F3b)? But other groups reported much better sol-gel HfO2-based RRAM. (Scientific Reports volume 9, Article number: 9983 (2019)). Please comment on that.

Reply:  Thank you very much for your comments. The high leakage of Au/HfO2/ITO device is attributed to the ITO substrate used, and similar results are shown on sol-gel prepared Ag/HfO2/ITO device [1]. Au is an inert metal electrode, and oxygen ions are stored at the interface of Au/HfO2. When a negative bias voltage is applied to Au, oxygen ions will move to the HfO2 layer. When the number of oxygen ions is too low, HfO2 becomes less insulated and HRS currents are high at this time, resulting in a small switch ratio. In the Cu/HfO2/p++Si devices mentioned in the literature, Cu is the active metal electrode, the conductive filament is composed of Cu ions, and the switching performance is related to Cu ions. Thanks again for your comment.

Comment4: Authors show the switching mechanism in F7. Now the question is in all switching is happening in HfO2 (F7d), then why do you need a complicated stack design? What is the purpose of each layer please explain.

Reply:  Thank you very much for your valuable comments and suggestion. We have corrected this error. Compared with HfO2 layer, Al-ZnO insertion layer can provide more oxygen vacancies and accelerate the formation of conductive filament. This can be seen in the Forming processes of Au/HfO2/ITO and Au/HfO2/Al-ZnO/HfO2/ITO device. HfO2/Al-ZnO/HfO2 device show an instantaneous increase in current, while pure HfO2 device show a slow increase in current. In addition, because the interface of different dielectric is the weakest point of the conductive filament, when the negative bias voltage is applied, the conductive filament will preferentially break at the HfO2/Al-ZnO interface. At the same time, oxygen ions stored at the HfO2/Al-ZnO interface will make the resistive layer more insulated. This results in a lower current of the HRS and the device thus obtains a higher switching ratio. Moreover, the conductive filament which is not completely dissolved in the HfO2 layer will become the starting point of the next conductive filament connection. This is analogous to a virtual electrode. The modified figure is shown in Figure 7 below. Thanks again for your comment.

Figure 7: A conductive filament model of a device

Comment5: As mentioned by the authors “In addition, in the durability test of the device, we could not get stable data, so the relevant durability of the device was not added in the manuscript.” You can not find the expected data always. But you must show what you have and explain it. There is always scope for improvement.

Reply:  We agree with the reviewer’s comment. According to your suggestion, we added the pulse endurance result of Au/HfO2/Al-ZnO/HfO2/ITO device, as shown in Figure S1. The HRS of the device fluctuated greatly, and after 120 pulse cycles, the HRS changed to LRS with resistive switching failure. The resistance fluctuation of the high resistance state is attributed to the presence of multiple conducting filaments inside the device. The resistive switching failure is due to insufficient oxygen ions recombination with oxygen vacancies. As a result, the conductive filament cannot be broken and the device remains at LRS. At the same time, this result will be added in the supplementary materials. Thanks again for your comment.

Figure S1: Pulse endurance of Au/HfO2/Al-ZnO/HfO2/ITO device.

Comment6: In the previous review, I mentioned that “The review of HfO2-based RRAM is very poor. Especially recent reports on the prospect and challenges of HfO2 are missing.” It is still not done. Reference 14 is almost 10 years old now. Try to include review papers (Small, 2107575, 2022) on HfO2 which will give a border view to readers.

Reply: Thanks for your constructive advice. According to your comments, we have added the application prospects and challenges of HfO2 in the manuscript. Thank you very much for your comments.

Thank you for your attentions. Your comments are helpful for our future work. 

Yours Sincerely,

 Yuan-Dong Xu

  1. Ramadoss, A.; Krishnamoorthy, K.; Kim, S.J. Resistive switching behaviors of HfO2thin films by sol–gel spin coating for nonvolatile memory applications. Applied Physics Express 2012, 5, 085803.

Round 3

Reviewer 1 Report

The only thing I like about the revised manuscript is that the authors supplied actual endurance data. I appreciate it. Yes, the endurance of the device can not be good and retention can not go for 85oC for 10 years which is necessary for memory application. But in the table please mention the retention temperature of each work. However, the other answers are not satisfying at all.

The author stated that “high leakage of Au/HfO2/ITO device is attributed to the ITO substrate used”, however, it is a wrong understanding. The leakage is coming from the HfO2 itself. Additionally, the better data of sol-gel HfO2 is not included in the reviewed manuscript. (Scientific Reports volume 9, Article number: 9983 (2019)). It shows either the authors are not able to understand the point, or they don’t want to improve the manuscript in a good way. It is not possible to highlight the problems in a single manuscript each and every time.

Authors mentioned that “Compared with HfO2 layer, Al-ZnO insertion layer can provide more oxygen vacancies and accelerate the formation of conductive filament.”. However, your forming data is suggesting a different story.

If the oxygen vacancy is high in the Al-ZnO structure the initial resistance will be low, then why the initial leakage current is high in HfO2? Look at your HRS and LRS for both devices (Figure 3b,c). Check HfO2 leakage behavior in other papers (Advanced Electronic Materials 7 (4), 2100022, 2021). Please read more HfO2-related papers.

How do you conclude that “the conductive filament will preferentially break at the HfO2/Al-ZnO interface”? Why at the top HfO2/Al-ZnO interface but not the bottom Al-ZnO/HfO2 interface? If AZO was deposited after the bottom HfO2 then how does the additional vacancy created in the upper HfO2? Again the mechanism is complete speculation without any actual experimental data support.

Author Response

Response Letter (nanomaterials-2042957)

Manuscript ID: nanomaterials-2042957

Title: Enhancement of Resistive Switching Performance in Hafnium Oxide (HfO2) Devices via Sol-gel Method Stacking Tri-layer HfO2/Al-ZnO/HfO2 Structures

Dear editors and reviewers,

Thank you very much for your insightful comments to our paper submitted to Nanomaterials. These comments are very helpful to revise the manuscript accordingly. Based on these comments, we have supplemented the relevant experiments and tests, added more sufficient explanations for the parts that need to be explained, and the corresponding modifications are shown in the manuscript. We reply to your comments and resubmitted to you the revised manuscript, we sincerely hope this revision can make this manuscript acceptable.

Below are the comments by you and our replies. All changes have been highlighted in red through the manuscript.

Comments from the editors and reviewers:

  •  

Referee 1:

Comments and Suggestions for Authors

The only thing I like about the revised manuscript is that the authors supplied actual endurance data. I appreciate it. Yes, the endurance of the device can not be good and retention can not go for 85oC for 10 years which is necessary for memory application. But in the table please mention the retention temperature of each work. However, the other answers are not satisfying at all.

The author stated that “high leakage of Au/HfO2/ITO device is attributed to the ITO substrate used”, however, it is a wrong understanding. The leakage is coming from the HfO2 itself. Additionally, the better data of sol-gel HfO2 is not included in the reviewed manuscript. (Scientific Reports volume 9, Article number: 9983 (2019)). It shows either the authors are not able to understand the point, or they don’t want to improve the manuscript in a good way. It is not possible to highlight the problems in a single manuscript each and every time.

Authors mentioned that “Compared with HfO2 layer, Al-ZnO insertion layer can provide more oxygen vacancies and accelerate the formation of conductive filament.”. However, your forming data is suggesting a different story.

If the oxygen vacancy is high in the Al-ZnO structure the initial resistance will be low, then why the initial leakage current is high in HfO2? Look at your HRS and LRS for both devices (Figure 3b,c). Check HfO2 leakage behavior in other papers (Advanced Electronic Materials 7 (4), 2100022, 2021). Please read more HfO2-related papers.

How do you conclude that “the conductive filament will preferentially break at the HfO2/Al-ZnO interface”? Why at the top HfO2/Al-ZnO interface but not the bottom Al-ZnO/HfO2 interface? If AZO was deposited after the bottom HfO2 then how does the additional vacancy created in the upper HfO2? Again the mechanism is complete speculation without any actual experimental data support.

Firstly, thank you very much for your valuable comments and suggestions. According to these comments, the modification but necessary changes have been made, which have been highlighted in red through the revised version and explained one by one as follows:

Reply: Thank you very much for your comments. According to your opinion, the test temperature of retention has been added in the table and the literature has been cited (Scientific Reports volume 9, Article number: 9983 (2019). (See Table 1 and Reference 39)

Leakage current: First of all, we admit that 'the high leakage current of HfO2 device is attributed to ITO bottom electrode' is a wrong understanding. Then, we supplement test the Forming process of another Au/HfO2/ITO device, as shown in Figure 1. It still shows no current surge and high leakage current. But unfortunately, in the past few days, we have not been able to find a reasonable explanation for this phenomenon. We are very sorry for this.

Figure 1: Forming process of Au/HfO2/ITO devices.

Model of conductive filament: First of all, we are sorry that we have no technical means to characterize where the conductive filament breaks, and we have not been able to prove the fracture location of the conductive filament through experimental means. Therefore, we analyze the conductive filament models of Au/HfO2/Al-ZnO/HfO2/ITO devices in combination with the conductive filament models of other double or triple layer devices[1-4]. The connection points of conductive filament of different sizes are weak. When a negative bias voltage is applied, oxygen ions stored at each interface recombine with oxygen vacancies. Therefore, there may be multiple rupture points in the conductive filament of the device. These rupture points are respectively at the interface of Au/HfO2, HfO2/Al-ZnO and Al-ZnO/HfO2, as shown in Figure 7d. Thanks again for your comments.

Finally, I know that such a reply is not satisfactory, but I can do nothing about it. Once again, I would like to express my apologies.

Figure 7: A conductive filament model for Au/HfO2/Al-ZnO/HfO2/ITO devices.

Thank you for your attentions. Your comments are helpful for our future work. 

Yours Sincerely,

 Yuan-Dong Xu

  1. Zhang, W.; Kong, J.-Z.; Cao, Z.-Y.; Li, A.-D.; Wang, L.-G.; Zhu, L.; Li, X.; Cao, Y.-Q.; Wu, D. Bipolar resistive switching characteristics of HfO2/TiO2/HfO2trilayer-structure RRAM devices on Pt and TiN-coated substrates fabricated by atomic layer deposition. Nanoscale Res. Lett. 2017, 12, 1-11.
  2. Lee, J.; Bourim, E.M.; Lee, W.; Park, J.; Jo, M.; Jung, S.; Shin, J.; Hwang, H. Effect of ZrOx/HfOxbilayer structure on switching uniformity and reliability in nonvolatile memory applications. Appl. Phys. Lett. 2010, 97, 172105.
  3. Wang, L.-G.; Qian, X.; Cao, Y.-Q.; Cao, Z.-Y.; Fang, G.-Y.; Li, A.-D.; Wu, D. Excellent resistive switching properties of atomic layer-deposited Al2O3/HfO2/Al2O3trilayer structures for non-volatile memory applications. Nanoscale Res. Lett. 2015, 10, 1-8.
  4. Zhang, W.; Lei, J.; Dai, Y.; Zhang, X.; Kang, L.; Peng, B.; Hu, F. Switching-behavior improvement in HfO2/ZnO bilayer memory devices by tailoring of interfacial and microstructural characteristics. Nanotechnology 2022, 33, 255703.
